# PGE2 Produced by Exogenous MSCs Promotes Immunoregulation in ARDS Induced by Highly Pathogenic Influenza A through Activation of the Wnt-β-Catenin Signaling Pathway

**DOI:** 10.3390/ijms24087299

**Published:** 2023-04-14

**Authors:** Resti Yudhawati, Kazufumi Shimizu

**Affiliations:** 1Department of Pulmonology and Respiratory Medicine, Faculty of Medicine, Universitas Airlangga—Dr. Soetomo General Academic Hospital, Surabaya 60286, Indonesia; 2Indonesia-Japan Collaborative Research Center for Emerging and Re-Emerging Infectious Diseases, Institute of Tropical Disease, Airlangga University, Surabaya 60286, Indonesia; 3Center for Infectious Diseases, Kobe University Graduate School of Medicine, Kobe 650-0017, Japan

**Keywords:** acute respiratory distress syndrome, mesenchymal stem cell, PGE2, β-catenin, influenza virus, infectious disease, NF-κB

## Abstract

Acute respiratory distress syndrome is an acute respiratory failure caused by cytokine storms; highly pathogenic influenza A virus infection can induce cytokine storms. The innate immune response is vital in this cytokine storm, acting by activating the transcription factor NF-κB. Tissue injury releases a danger-associated molecular pattern that provides positive feedback for NF-κB activation. Exogenous mesenchymal stem cells can also modulate immune responses by producing potent immunosuppressive substances, such as prostaglandin E2. Prostaglandin E2 is a critical mediator that regulates various physiological and pathological processes through autocrine or paracrine mechanisms. Activation of prostaglandin E2 results in the accumulation of unphosphorylated β-catenin in the cytoplasm, which subsequently reaches the nucleus to inhibit the transcription factor NF-κB. The inhibition of NF-κB by β-catenin is a mechanism that reduces inflammation.

## 1. Introduction

Acute respiratory distress syndrome (ARDS) is an acute respiratory failure that arises due to acute lung injury which causes a complication [1,2]. ARDS is characterized by diffuse alveolar damage, which leads to many complications, ultimately severe hypoxemia and respiratory failure [1,3]. The etiology of ARDS is quite complex; one factor is bacterial- and viral-induced pneumonia [4,5]. Influenza A virus is one of the reasons for the high incidence of ARDS and fulminant ARDS (FARDS), which includes the highly pathogenic avian influenza (HPAI) H5N1 [6,7]. On 23 February 2023, the Cambodia International Health Regulations (IHR) National Focal Point (NFP) notified WHO of a confirmed case of death of a girl with severe pneumonia induced by avian influenza A (H5N1) virus [8]. In addition, SARS-CoV-2 is another infectious virus reported to induce ARDS [9,10,11].

The mortality of ARDS remains high, although there has been an overall improvement in diagnosis and supportive care. However, no pharmacological therapy is available [12]. According to a meta-analysis study, antiviral medicine is effective for the earliest infection but has little beneficial effect on severe influenza [13]. This medicine cannot repair the inflammation related to associated tissue injury induced by the influenza virus [14]. Developing treatments prioritizing immunomodulation and inhibiting the intracellular signaling cascade, as well as limiting viral replication, is essential because any patient infected with the highly pathogenic influenza virus already has severe complications at admission. Cell-based therapy is a potential therapeutical approach. Allotransplantation therapy utilizing mesenchymal stem cells (MSCs) has shown beneficial effects on ARDS [15]. Furthermore, it has negligible immunogenicity features [16]. Other studies on MSCs have demonstrated their therapeutic potency on multiple organ injuries, including myocardial infarction [17], diabetes [18], sepsis [19], liver diseases [20], and acute kidney failure [21]. In an experimental model of acute lung injury, either due to chemical, viral, or bacterial infection, MSCs were found to remain beneficial [22,23,24]. In addition, studies on MSCs administration to acute lung-injury-virus-induced ARDS are still limited. Studies on MSCs’ effects on lung injury induced by the H1N1 virus have shown conflicting results [25,26]. In contrast, studies of highly pathogenic influenza A virus subtypes H9N2 and H5N1 have indicated that MSCs are effective on lung-injury-induced ARDS [27,28]. However, the mechanism that leads to MSCs mobilization to the damaged tissue site as an immunomodulator remains unclear.

Most studies at present focus on investigations of the regeneration effect of MSCs and its application in treatments that aim to improve tissue injury and regeneration. Knowledge about the crosstalk between regulatory factors in MSCs niches is essential, as it functions in the homeostasis of normal and injured tissue (such as that affected by infection). This pathway is extensively involved in the regulation of adult MSCs’ fate in various tissues. Through the canonical pathway, this Wnt/β-catenin pathway helps the maintenance of MSCs and improves their proliferation ability as a function of regeneration [29,30,31]. However, the role of Wnt signaling in the immunoregulatory function has not drawn much attention as yet. In this review, we seek to address the mechanisms of MSCs as immunomodulatory agents through the canonical Wnt/β catenin signaling pathway.

## 2. Host Immune Response and Cellular Signaling Pathway Activation Induced by Influenza Virus

The influenza virus is a member of the Orthomyxoviridae family, an enveloped corona virus with a diameter of approximately 80–120 nm. The genome is single-stranded ribonucleic acid (RNA) with a negative sense, consisting of eight (type A and B) or seven (type C) segments. Hundreds of spikes protrude from the surface of the type A virus’s envelope, e.g., hemagglutinin (HA) and neuraminidase (NA). HA is an essential glycoprotein antigen, acting as the site for viral binding and fusion between viral and cell membranes. NA possesses an enzymatic activity, acting to destroy the receptor molecules preventing virus aggregation and releasing the newly assembled virion of the infected cell [32].

The characteristic innate immune response is the primary determinant of the outcome of influenza infection [33]. Three main classes of pattern-recognition receptors (PRRs) recognize the pathogen-associated molecular patterns (PAMPs) of the influenza A virus, e.g., cytosolic retinoic acid inducible gene 1 (RIG-1) sensor [34], toll-like receptor (TLR) 7 in the endosome [35], and nucleotide-binding domain-leucine-rich repeat-containing (NLR) molecules [36]. These three pathways can induce pro-inflammatory cytokines and chemokines, which leads to pathological lung damage [33,36]. The host signaling pathway is directed towards the influenza virus after the virus binds with the host through HA and NA proteins. The HA protein can directly induce the expression of several host cytokines, including TNFα and IL-1 [37]. Host receptors can recognize ribonucleocapsid protein (RNP) and initiate several signaling transductions. The resulting transduction signaling pathways are complicated and overlap through the PKR, ASK1, TLR, RIG, phosphoinositide 3-kinase (PI3K), and PKC pathways. PKR plays roles in the apoptosis processes of cells infected by the influenza A virus through Fas and ligand induction. PKR activation can also activate IKK, which can lead to the phosphorylation and degradation of Iκβ (inhibitory kappa beta), and then cause the active transcription factor, nuclear factor kappa B (NF-κB) subunit p65/p50, to release. NF-κB subunit p65/p50 can induce pro-inflammatory cytokines, chemokines, and proapoptotic molecules, such as INFβ [38].

The single-stranded RNA (ssRNA) of the influenza A virus is recognized by TLR 7 and 8 via the TLR pathway; it then induces several adaptor molecules and transcription factors, including MyD88, TAK1, TAB1, TAB2, and TRAF6, and activates NF-κB. The interaction between viral protein and TLR activates IKK, which can lead to the phosphorylation and degradation of Iκβ, then release the NF-κB transcription factor subunit p65/p50. Through TKR7/8, TANK binding kinase (TBK) 1 and IKK may also be activated, which then induces the activation of the NF-κB transcription factor and interferon regulatory factor (IRF) 3/7, which expresses the interferon β gene and apoptosis regulation. Influenza A can also be recognized by RIG 1, MDA 5 (melanoma differentiation-associated gene 5), and LGP2. RIG1 and MDA5 act as cytoplasmic viral RNA sensors that play an essential role in innate antiviral immunity [37,38].

Viral proteins synthesized in the infected cells partially degrade into smaller peptides in the endosomes/lysosomes, and they form a complex with the MHC class I or II peptides at the cell surface. CD4+ helper T cells recognize the complex with MHC class II [32,33], whereas CD8+ cytotoxic T cells recognize the complex with MHC class I. Such antigen presentations and recognition by T cells activate the adaptive immune response.

## 3. Cytokine Storm and ALI/ARDS Induced by Influenza Virus Infection

Pro-inflammatory cytokines and chemokines induced by influenza virus infection cause pathological lung damage [33,36]. Highly pathogenic influenza A virus infection causes the release of excessive cytokines (cytokine storm). The release of this large amount of cytokines initially suppresses viral replication. However, this cytokine storm causes local cell death and severe damage to lung tissue, and eventually results in fibrosis. In the early stages, viral pneumonia occurs in interstitial pneumonitis. This process continues with exudation, intra-alveolar edema, inflammatory cells, and erythrocytes mobilization from the surrounding capillaries, as well as hyalin membrane formation and fibroblast accumulation. The inflammatory cells then produce many inflammatory mediator cells, which are clinically referred to as ARDS. As a result, the disruption of oxygen diffusion occurs, so hypoxia/anoxia can damage other organs (anoxic multiorgan dysfunction) [39].

The mechanisms of cytokine storm damage proceed through multiple stages. When the highly pathogenic influenza A virus infects alveolar epithelial cells, these cells release various pro-inflammatory cytokines and chemoattractants. HPAI and their infected cells can be recognized by pattern recognition receptors (PRRs) on innate immune cells, such as airway epithelial cells, inflammatory DCs, macrophages, monocytes, and neutrophils. There are several types of cellular PRRs, as follows: membrane-associated toll-like receptors (TLRs), C-type lectin receptors, cytoplasmic nucleotide-binding oligomerization domain-like receptors, and retinoic acid induced gene-like receptors [40]. The viral RNA in the endosome and cytoplasm is predominantly recognized by membrane-associated TLRs and RIG-I-like receptors, respectively. PRRs recognize pathogen-associated molecular patterns (PAMPs) on different types of microorganisms [41,42]. The engagement of PRRs by PAMPs on HPAI activates PPR-related signaling and induces the expression of various pro-inflammatory cytokines and chemokines to recruit innate immune cells (such as macrophages, neutrophils, and DCs) into the infected airway [43]. Macrophages present viral peptides on the cell surface and activate lymphocyte T cells. Activated macrophages as well as lymphocyte T cells then release various pro-inflammatory cytokines and chemoattractants. All chemokines and cytokines can recruit more macrophages and neutrophils to the lung and, thus, cause more extensive damage [44].

The TLRs and RIG-I-like receptor on the cellular membrane can interact with their specific extra-cellular and endosomal PAMPs [43]. TLR7 signaling, via the adaptor protein MyD88, activates IRF7 and NF-κB, while RIG-I signaling via the adaptor molecule IPD-I located in the mitochondria can activate IRF3 and NF-κB [36,45]. The activation of IRF-7 and IRF-3 may cause IRF-7 and IRF-3 translocation into the nucleus and produce interferon type I, while NF-κB acts as a transcription factor to induce pro-inflammatory cytokine production, including IL-6, TNFα, and pro-IL-1β [36,45]. 

In addition to PAMP, DAMP (danger-associated molecular pattern) is a host molecule that functions to maintain PRRs’ activation. High-mobility group box 1 (HMGB1) is a DAMP released from damaged cells due to ARDS [46,47]. HMGB1 has high affinity in the RAGE ligand, and its regulation is increased in mice with pneumonia induced by influenza virus infection [48]. The binding of HMGB1 with RAGE produces a signal that activates NF-κB then induces adhesion molecules and pro-inflammatory cytokines [48,49]. Figure 1 illustrates the host immune response and cellular signaling pathway activation induced by the influenza virus, as well as the mechanisms of the cytokine storm.

## 4. Mesenchymal Stem Cells

MSCs are non-hematopoietic stem cells with the ability to differentiate into mesenchymal and non-mesenchymal lineages, and they originate from mesodermal tissues. These cells can be found in all human organs, specifically in the regions of cell populations found in the perivascular area, mainly in bone marrow, the umbilical cord (blood or Whartons jelly), and fatty tissues. MSCs in the lung can differentiate into the epithelial phenotype and structure lung cells [50]. Currently, autologous and allogeneic MSCs are being investigated in clinical research for the treatment of various diseases, including diabetes mellitus, multiple sclerosis, Crohn’s disease, and end-stage liver disease, as well as to restore left ventricular function in congestive heart failure patients and prevent transplant rejection.

MSCs, sometimes also referred to as marrow stromal stem cells, were first identified in the guinea pig by Friedenstein in 1968 [51]. The first human MSCs were isolated by Haynesworth and Caplan in 1991 [52]. Specific cell surface markers for MSCs are unavailable; hence, the International Society of Cellular Therapy in 2006 defined that MSCs must meet the following three criteria: (1) Under standard tissue culture conditions, MSCs must grow while attached. (2) They must express specific cell surfaces markers, such as CD73, CD90, and CD105, but not express the hematopoietic markers CD11b, CD14, CD34, or CD45. 3) They must have the capacity to differentiate under in vitro conditions into mesenchymal lineages, including adipocytes, osteoblasts, and chondroblasts [53]. These characteristics of isolated human MSCs were all originally described by Pittenger et al. [54].

MSCs are multipotent cells that can differentiate into mesodermal, endodermal, and ectodermal cells [55]. MSCs also induce hematopoietic cells to release several regulatory factors, such as growth factors and anti-inflammatory cytokines, which can modulate the immune response [56]. MSCs were first introduced as an adherent to a clonogenic, non-phagocytic, fibroblastic cell population. Furthermore, these cells can be isolated from various other tissues, including adipose tissue, umbilical cord, placenta, and amniotic fluid [55].

### 4.1. Allogeneic MSCs

Allogeneic MSCs are exogenous stem cells from the same species as MSCs, but they are not genetically identical. The advantages of allogeneic MSCs (allo-MSCs) in therapeutic applications are manifold compared with autologous MSCs (auto-MSCs). Moreover, auto-MSCs have several associated limitations, including the difficulty of obtaining adequate auto-MSCs from some patients, such as light-weight patients or patients with myelofibrosis. Auto-MSCs isolated from donors of advanced age show decreased biological activity, including activity regarding their differentiation and regenerative potential [57]. Some systemic illnesses, such as diabetes, can affect the inherent features of auto-MSCs, hence reducing their protective effect. Furthermore, it is challenging to collect sufficient amounts of healthy auto-MSCs with high activity from individuals with this condition [58]. The first study seeking to mix MSCs with isolated immune cells—T cells, B cells, and NK cells—and show that each responded to MSCs was undertaken by Aggarwar and Pitternger [59], and this was also the first example of the identification of PGE2 as an effector released from MSCs.

Some other limitations of using auto-MSCs include the very time-consuming nature of auto-MSCs extraction, meaning it is hard to use them quickly against acute diseases such as stroke and myocardial infarction. Meanwhile, allo-MSCs are easily obtainable, allowing for immediate administration. As a result, using allo-MSCs represents a promising alternative therapy to auto-MSCs, with advantages in terms of time and cost efficiency, as well as quality assurance [60]. The activity of various immune cells, including dendritic cells, NK cells, B cells, and T cells, through cell-to-cell contact and soluble factors can be inhibited by MSCs [61]. Allo-MSCs also contribute to lowering the immune response after implantation, owing to their low immunosuppressive and immunogenicity properties.

MSCs exhibit low levels of MHC class I molecules on their surface and lack expressions of MHC class II as well as the co-stimulatory molecules CD80 (B7-1), CD86 (B7-2), and CD40, which may limit immune recognition. However, under infection conditions during which pro-inflammatory factors are released and inflammation occurs, the expression of MHC-II can be up-regulated in antigen-presenting cells (APCs) [62]. Klyushnenkova et al. supported this statement after demonstrating that MSCs treated with IFN-γ showed up-regulated MHC-II molecules [63]. Additionally, MSCs under inflammatory circumstances show an up-regulation of the co-stimulatory molecule CD40. However, T cells are not fully activated because CD40 expression is counterbalanced by the inhibitory molecule CD274 (also known as PD-L1), which is also expressed by MSCs [64]. The hypothesis that allo-MSCs have the same efficacy as auto-MSCs has been proven. However, in vivo studies have revealed that allo-MSCs are not completely immunogenic, thus possibly eliciting an immune response despite their low immunosuppressive and immunogenicity. In vivo and in vitro studies have both reported these findings. However, until recently, this has been a point of contention, with inconclusive results [65,66].

### 4.2. MSCs Mode and Delivery

There are several techniques for MSCs administration, depending on the underlying clinical indications and pathological abnormalities. The proper selection of the mode of delivering MSCs provides optimal therapeutic efficacy [67]. There are two main methods used to deliver cells into the body: local delivery into the tissue, and systemic delivery. Local delivery is via injection, for example intraperitoneal (IP), intramuscular (IM), or intracardiac, whereas systemic delivery methods involve the administration of cells intravenously (IV) or via intra-arterial (IA) paths [68]. The alternative route of delivery is direct injection (DI) into the tissue or organ targeted in order to escalate the engraftment and/or local differentiation [69].

Another method commonly used to deliver MSCs is via intravascular (IV) infusion, owing to its ease of utilization and low risk [70]. However, cells administered through IV have to pass through the lungs first before being distributed to the rest of the body. This condition leads to a substantial problem called the pulmonary “first-pass” effect, which produces high numbers of entrapment cells [71]. This situation occurs because MSCs are approximately 20–30 µm in diameter, and most particles of this size are filtered out by the lungs [71]. Nevertheless, MSCs entrapment in the lungs provides therapeutic effects on the targeted tissues. A concise review study concluded that MSCs isolated from bone marrow alter the tissue microenvironment by secreting soluble factors (autocrine–paracrine). This makes a major contribution to tissue improvement via the trans-differentiation capacity [72]. Several studies have proven that engraftment plays an important role in MSCs therapy aimed at injury; however, this engraftment ability may not play a key therapeutic role in the lungs. Because lungs’ engraftment ability is less than 5%, the paracrine mechanisms play greater roles [73,74]. The beneficial effects of MSCs are derived from their capacity to secrete paracrine factors that modulate immune response and alter the response of the epithelium, endothelium, and inflammation cells to injury. Administration via IV has been extensively reported to confer therapeutic effects through mechanisms involving secondary signaling effector cell systems and interactions with the host immune systems [69].

Theoretically, the intra-arterial (IA) delivery route enables cells to pass through the lungs at least once, thus avoiding the pulmonary “first-pass” effect, and consequently more cells are able to reach the targeted tissues [68].

Several studies have reported the positive therapeutic effects of MSCs; however, there are still a lot of questions and challenges related to the application of MSCs. Cell therapy can induce cell emboli, which may potentially increase mortality. This type of embolism depends on the concentrations and rates of cell administration [69,75]. Other risks may arise, such as MSCs transformation, potential adverse inflammatory effects, tumor formation, or thrombosis associated with the intravenous infusion of MSCs [70]. Until recently, the safety of MSCs therapy has been under question, but the efficacy and the resulting interaction in the microenvironment of the host remain controversial.

### 4.3. The Mechanisms of Paracrine, Endocrine, and Cell Engraftment of MSCs

The mechanisms of MSCs as immunomodulatory, improving and regenerating cells, still need to be explored. Some articles have mentioned that paracrine mechanisms are enacted through the secretion of soluble mediators and cell-to-cell contact. Cell-to-cell contact depends on paracrine mechanisms in cells with limited differentiation potential, such as MSCs [76]. The soluble factors include interleukin-1 receptor antagonist [77], interleukin-10 [78,79], prostaglandin E2 (PGE2) [78], LL-37 [80], keratinocyte growth factor [81], and angiopoietin-1 [82].

A study conducted by Ortiz et al. [83] suggested the activity of paracrine mechanisms enacted via soluble factors rather than trans-differentiation mechanisms. Other studies have stated that the role of trans-differentiation is less significant than that of paracrine effects caused by the release of soluble factors in response to acute lung damage [84]. Moreover, the capacity for lung engraftment is less than 5%, and, thus, the paracrine mechanism plays a significant role [73,74].

Krause et al. [85] proved that bone-marrow-derived cells can produce cells from several different organs, including the lungs. Preliminary studies suggest that engraftment plays a vital role in treating injury by MSCs. However, the engraftment properties of the lung do not play a significant therapeutic role. Instead, the capacity of MSCs to secrete paracrine factors is beneficial in modulating immune responses and in altering the responses of the endothelium or epithelia and inflammation cells toward injury [62,86,87].

### 4.4. MSCs and Niche Inflammation

In addition to the repair and regeneration activities of MSCs in microenvironments where inflammation occurs (inflammation niche), evidence suggests that MSCs have immunomodulatory properties which help in maintaining the immature state of dendritic cells (DCs) through the inhibition of MHC class II, CD1a, CD40, CD80, and CD86 expression and the suppression of pro-inflammatory cytokines’ production [88]. An in vitro study showed that the ability of MSCs to secrete PGE can be increased by altering their environmental conditions. The study showed that exposure to hypoxia can stimulate angiogenic and antiapoptotic factors, such as VEGF, FGF-2, HGF, and IGF-1. MSCs’ exposure to IFγ and TNf-α in culture increased the expressions of PGE2, IDO, TGFβ, and HGF [89].

The immunosuppressive effect produced by MSCs is influenced by several conditions, such as the source of the isolated MSCs, the number of passages in the culture prior to use, the MSC dosages, and the recipient’s specific pathological conditions; however, the immunomodulatory roles of MSCs have attracted great interest on the part of researchers and clinicians [90].

### 4.5. The Immunobiology and Communication between MSCs and Damaged Tissue

MSCs in situ play an essential role in the cellular homeostasis of host tissue by replacing dead or dysfunctional cells. The capacity of MSCs to modulate immune responses has been shown in an in vitro study [91]. Tissue injury is often accompanied by inflammation, and the inflammation factors may cue a signal that mobilizes MSCs to the damaged tissue. Before carrying out the repair function, the initial preparation of the microenvironment is required. 

Tissue damage is commonly accompanied by the activation of immunological or inflammatory cells. The factors produced by cells undergoing apoptosis, necrosis, microvasculature damage, and stroma recruit not only macrophages and neutrophils but also adaptive immune cells, such as CD4 T lymphocytes, CD8 T lymphocytes, and B cells [91,92]. Meanwhile, phagocytic cells release inflammatory mediators, such as TNFα, IL-1β, chemokines, free radicals, and leukotrienes, in response to cell damage. Thus, inflammatory chemicals and immune cells, fibroblasts, and endothelial cells regulate changes in the microenvironment, and as a result, MSCs mobilize and develop, stimulating the replacement of damaged tissue cells. MSCs may originate either from tissue-resident cells or bone marrow. However, the processes of the mobilization of MSCs to damaged tissue sites are poorly understood. Moreover, the mechanisms of how MSCs survive and differentiate into different cell types remain unclear [90].

Cytokines, such as TNFα, IL-1β, and IFNγ, toxins from infectious agents, or hypoxia can induce the release of many MSCs growth factors, such as fibroblast growth factor (FGF), epidermal growth factor (EGF), platelet-derived growth factor (PDGF), keratinocyte growth factor (KGF), vascular endothelial growth factor (VEGF), hepatocyte growth factor (HGF), transforming growth factor-β (TGF-β), insulin growth factor-1 (IGF-1), angiopoietin-1 (Ang-1), and stromal-cell-derived factor-1 (SDF-1), all of which are induced once the MScs enter the microenvironment [90,93]. The development of fibroblasts, endothelial cells, and tissue progenitor cells is then promoted by these growth factors, which is essential to tissue regeneration and repair [90].

### 4.6. The Immunosuppression Property of MSCs

The inflammatory niche determines the ability of MSCs to inhibit immune cell activity. IFNγ, together with pro-inflammatory cytokines, such as IL-1α, TNFα and IL-1β, the occurrence of a chemokine burst, and the expression of adhesion molecules, including intercellular adhesion molecule-1 (ICAM-1), CCR5 ligand, CXCR3 ligand, and vascular cell adhesion molecule-1 (VCAM-1), can stimulate MSCs to release large amounts of immunosuppressive factors [94]. The co-activation of these inflammatory molecules induces the accumulation of immune cells near to MSCs; these microenvironmental conditions stimulate MSCs into potent immunosuppressive behavior. Immunosuppressive factors, such as PGE2, IL-10, IL-6, LIF, TSG6, HO-1, and CCL2, can also affect immune cell activation [90]. The communication between the MSCs in damaged tissue and the immunosuppressive property of MSCs can be seen in Figure 2. PGE2 is an immunosuppressive factor with roles and mechanisms that should be investigated comprehensively in the future.

### 4.7. The Immunomodulation Roles of Exogenous MSCs in Lung Injury

MSCs have unique immunomodulatory and paracrine properties that are advantageous in treating diseases associated with inflammation and organ injury [95]. Several studies have proven that MSCs enact immunosuppressive effects by inhibiting the activity of innate and adaptive immune cells. Cell-to-cell contact mechanisms mediate the immunosuppressive effects. In vitro studies have shown that the PGE2 derived from murine and human MSCs acts as an immunosuppressant by inducing Foxp3þTregs [96], inhibiting NK cells [97], inducing M2 cells [98], and inhibiting dendritic cell maturation [99]. Research on sepsis models has proven that MSCs derived from bone marrow cells can improve sepsis via immunosuppression mechanisms through PGE2. PGE2 also reprograms the host macrophages to increase the production of IL-10, an anti-inflammatory cytokine [78]. Research by Sun et al. [100] administered intratracheal MSCs 4 h after lung injury on mice models with lipopolysaccharide (LPS)-induced ALI and demonstrated that alveolar levels of CD4+CD25+Foxp3+ Treg were reduced and the balance of anti-inflammatory and pro-inflammatory cytokines was maintained. The study by Mei et al. [101] proved that the administration of MSCs intravenously 6 h and 24 h after polymicrobial lung injury in mice models reduced acute inflammatory responses and increased phagocytosis via bacterial clearance.

The study of Ortiz et al. [77] proved that the intravenous administration of MSCs immediately after lung injury in a rat model due to the intratracheal administration of bleomycin could induce the secretion of IL-1 receptor antagonists, prevent macrophages from producing TNFα, and inhibit IL-1α secretion by T lymphocyte cells. In contrast, Rojas et al. [74] proved that the intravenous administration of bone-marrow-derived MSCs in mice models 6 h after lung injury caused by intratracheal administration of bleomycin increased G-CSF and GM-CST levels in circulation and reduced inflammatory cytokines. Furthermore, Maron-Gutierrez et al. [102] demonstrated that intravenous MSC administration to a murine model with ARDS one day after LPS induction reduced inflammation, inhibited fibrogenesis, increased MMP-8 expression, decreased TIMP-1 expression, and caused a shift in macrophage phenotype from M1 to M2.

Several studies used MSCs that were not derived from bone marrow, including Shin et al. [103], with outcomes suggesting that the intravenous administration of human adult stem cells from adipose tissue 30 min after the induction of LPS in rat models with endotoxemia decreased the level of inflammatory cytokines in serum and lungs and prevented renal apoptosis and repaired multiorgan injury. Zhang et al. [104] proved that administering human stem cells and mouse stem cells derived from adipose tissue in mouse models with LPS-induced ARDS increased IL-10 levels and reduced neutrophil levels, lung permeability, and pro-inflammatory cytokine production. Furthermore, Li et al. [28] proved that the intravenous administration of MSCs derived from the human umbilical cord 1 h after LPS induction in rat models with ARDS could reduce mortality, as well as serum TNFα, IL-1β, and IL-6, without reducing IL-10 levels. Chang et al. [105] demonstrated that the intratracheal administration of stem cells derived from the umbilical cord in rat models on postnatal Days 3 and 10 showed a significant protective effect only at the beginning of the inflammation. Several publications evaluating the success of non-mesenchymal-cell-based therapy in ARDS, including that by Ornellas et al. [106], have proven that the administration of BMDMC reduced inflammation and remodeling in mouse models with ARDS due to sepsis. Nandra et al. [107] also suggested that the intravenous administration of BMDMCs in rat models with hemorrhagic shock could reduce kidney and liver damage through the enhancement of PKB and glycogen synthase kinase (GSK)-3 phosphorylation, the weakening of NF-κB activation, and ERK1/2 and ICAM-1 phosphorylation.

A study by Gorman et al. [108] demonstrated that in mechanically ventilated patients with moderate to severe ARDS, increasing doses (100, 200, or 400 × 106 cells) of a single intravenous infusion of human umbilical-cord-derived MSCs (ORBCEL-C) were well-tolerated, supporting the safety of the intravenous administration of ORBCEL-C. However, no important trends over time (from baseline to Days 4, 7, and 14) were identified, and there were no apparent dose-dependent effects of MSCs on any of the plasma markers of systemic inflammation (IL-6, IL-8, or IL-18), epithelial cell injury (SP-D), or endothelial injury/activation (Ang-2/ICAM-1).

Dilogo et al. [23] investigated the intravenous administration of umbilical cord mesenchymal stromal cells in critical patients with COVID-19. They revealed reductions in IL-6 in the MSCs group and an IL-10 increment on Day 7 after MSC application in comparison to baseline. The study also showed the increased expression of LIF (leukemia inhibitory factor) in 80% of recovered patients who were treated with MSCs. LIF shows the ability to repair and regenerate through stem cell niches of type II alveolar epithelial cells. They also saw the suppression of the populations of CD8-CXCR3 and CD56-CXCR3 after MSC application. VEGF also showed an increasing trend in the MSCs group, compared with a decreasing trend in the control group on Day 7. VEGF is an angiogenic factor that is essential in the recovery of damaged lungs, which must be introduced into circulation so that the regeneration of lung capillaries can occur.

Adas et al. [109] suggested that administering MSCs to critically ill COVID-19 patients decreased serum ferritin, IL-6, fibrinogen, and CRP levels significantly. The levels of other pro-inflammatory cytokines (IFNγ, IL-2, IL-12, and IL-17A) appeared to steadily decrease in the MSC transplanted group. The anti-inflammatory cytokines IL-10, IL-13, and IL-1ra appeared to increase steadily and statistically significantly from baseline.

### 4.8. The Effect of Exogenous Mesenchymal Stem Cells on Influenza-Virus-Induced Lung Injury

MSCs’ administration to ARDS induced by influenza A viral infection has yet to be widely studied; however, investigation on humans is still limited. Several studies focusing on different sub-types of the influenza virus have shown beneficial results. Table 1 summarizes the research and the viruses used, the sources of the MSCs administered, and the results. 

## 5. Prostaglandin E2 Acts as an Immunosuppressive Properties of Mesenchymal Stromal Cells

Depending on the stimulation in the microenvironment, MSCs can act in both immune activation and immune suppression. These mechanisms are also known as sensory functions and “switchers” of the body’s immune system [116]. PGE2 plays a key role in the immunosuppressive properties of MSCs. An in vitro study demonstrated that MSCs exposed to IFN-γ and TNF-α in a culture of MSCs increased the expression of PGE2 [89]. MSCs constitutively secrete the soluble factors of PGE2 without the activation signal. In addition, PGE2 is easily detected in the culture very quickly in only 4 h [117].

Li et al. suggested that MSCs transfected to overexpress cyclooxygenase (COX)-2 is a potent immunomodulator [118]. Daniel et al. [117] also confirmed that PGE2 indicates the therapeutic efficacy of mesenchymal stem cells; they proved that exogenous PGE2 is sufficient to inhibit the secretion of TNF-α and IFN-γ. PGE2 not only represents the crucial immunomodulator factor expressed by MSCs but also functions as a marker to assess the efficacy of MSC therapy. A study by Yañez et al. [119] suggested that myeloid dendritic cells (m-DCs) co-cultured with Ad-MSCs or BM-MSCs notably increased the level of PGE2 detected in the culture of MSCs.

As an essential mediator, PGE2 modulates many biological functions, including immune responses, blood pressure, gastrointestinal integrity, and fertility. PGE2 dysregulation or degradation has been linked to a number of pathological conditions [120]. PGE2 is a prostaglandin produced from arachidonic acid (AA) by COX. Many cell types, including macrophages and dendritic cells, can produce COX-derived PGE2 [121]. PGE2 is an essential mediator that modulates various physiological and pathophysiological processes through autocrine or paracrine mechanisms. PGE2 is mediated by four G-protein-coupled receptors (EP1-EP4), each of which has a unique physiological and pathophysiological effect [121]. PGE2 is involved in the proliferation and migration of several cell types, in addition to its role in the inflammatory response [122,123].

Some evidence suggests the beneficial effects of PGE in the lung, and its role in several physiological processes, such as the maintenance of barrier integrity [124,125]. Several studies have shown that PGE2 may trigger cell proliferation and inhibit apoptosis in lung epithelial cells [124,125]. COX is a key enzyme in arachidonic acid’s (AA) conversion into prostaglandin. COX can maintain and regulate the physiological functions of blood vessels, kidneys, and other organs by producing substances such as PGE2. Some studies have found that PGE/COX-2 is closely correlated with the Wnt/β-catenin signaling pathway. An in vitro study proved that PGE2 activation enables unphosphorylated β-catenin to accumulate in the cytoplasm, then get into the nucleus and activate genes involved in regeneration functions [122,126,127,128]. Another study by Zheng et al. [129] demonstrated that PGE2/COX-2 was involved in fracture healing by promoting the transcription and translation of the key gene of β-catenin in the Wnt/β-catenin signaling pathway; this pathway facilitates the formation of calluses and osteoblast proliferation and accelerates fracture healing.

The activation of the β-catenin pathway not only functions in the regeneration process but also is active in immunoregulation mechanisms. According to several studies, the mechanisms of immunoregulation act via the cross-regulation between the Wnt/β-catenin and NF-κB signaling pathways [130,131,132,133,134].

## 6. PGE2 Promotes Immunoregulation through Activation of Wnt-β-Catenin Signaling

PGE2, produced via the COX pathway, has three identified COX isoforms. One is COX-2, which is undetectable in most tissues but is activated in response to various inflammatory mediators, including pro-inflammatory cytokines. COX-2 is associated with the pathogenesis of several diseases [135,136]. The literature mentions that in damaged tissue, PGE2 is increased as a function of the homeostasis mechanism [124].

The β-catenin protein is a subunit of the cadherin protein complex that acts as an intracellular signal transducer, which is also involved in the regeneration and immunoregulation process. β-catenin is a double-function protein that regulates the coordination of cell adhesion and the transcription of genes. β-catenin is homologue with γ-catenin, also known as plakoglobin, and is expressed in various tissues [137]. The Wnt/β-catenin signaling pathway is a crucial pathway for several biological processes. The aberration of this signaling can be found in several human diseases [138].

Wnt/β-catenin signaling also acts as the regulation pathway of the stem cells niche in several tissues. However, the role of Wnt signaling in the immunoregulatory function has not been extensively reported, although it is broadly involved in the regulation of adult MSCs’s fate in various tissues. Through the canonical pathway, this Wnt/β-catenin pathway helps the maintenance of MSCs and improves their proliferation ability as a function of regeneration [29].

The available studies suggest that one of the main roles of Wnt signaling is determining the fate of and controlling stem cells [139]. Gehart and Clevers [140] reported one of the main communicative roles in the stem cells niche is played by the Wnt/β-catenin pathway. The activation of the canonical Wnt pathway, which is dependent on β-catenin, is essential to the biology of MSCs and induces differentiation and proliferation [139].

The elevation of β-catenin under certain tissue damage conditions is indispensable. During viral infection, different transduction signaling pathways control various gene expressions that regulate several molecular and cellular activities required for the eradication of microorganisms and the regulation of inflammation [134]. The inflammation response must be regulated tightly because uncontrolled inflammation can lead to tissue damage. Hence, the β-catenin signaling pathway is crucial due to its role in inhibiting the expression of several inflammation molecules during pathogen infection. In addition, β-catenin signaling also plays an essential role in the processes of cell differentiation and stem cell self-renewal maintenance [141].

### 6.1. β-Catenin Protein Activates Wnt-β-Catenin Signaling

β-catenin is a CTNNB1 protein encoded by the human CTNNB1 gene that regulates the coordination of adhesion between cells and gene transcription. This protein acts as the main signal transmitter in the Wnt signaling pathway. β-catenin is also detected in adherent junctions, forming a complex with E cadherin and actin filaments from the cytoskeleton [142].

Wnt/β-catenin is a fundamental pathway essential for embryogenesis, cell differentiation, and the maintenance of stem cell self-renewal in humans, lime molds, and worms. Because of the destruction of the complex controlling the level of free β-catenin (which refers to the form unbound to the E-cadherin complex), β-catenin is undetectable in the cytoplasm, nucleus, or unstimulated cells. This complex is made up mainly of the tumor suppressor adenomatous polyposis coli (APC) and Axin, which serves as a protein scaffold for the Ser/Thr protein kinase (casein kinase 1α (CK1α)) and GSK-3β to phosphorylate β-catenin at the N-terminal residues Ser45, Ser33, Ser37, and Thr41. This phosphorylation labels β-catenin, enabling it to be polyubiquitinated by the Skp1-Cdc53-F-Box E3 ubiquitin ligase complex (SCFβ–TRCP) and degraded by the 26S proteasome. GSK-3α is able to regulate β-catenin phosphorylation both in vitro and in vivo, suggesting that both isoforms of GSK-3 contribute to maintaining low levels of β-catenin under baseline conditions [134].

#### 6.1.1. Wnt Signaling and Regulation

Axin molecules oligomerize with each other in resting cells via the C-terminal DIX domain, which has two interfaces that bind together, forming linear oligomers or even polymers in the cell’s cytoplasm. The only other protein known to have a DIX domain is Dishevelled, which has unique characteristics. A single Dsh protein from Drosophila is related to three paralogous genes in mammals, namely Dvl1, Dvl2, and Dvl3. It corresponds to the cytoplasmic region of the receptor frizzled with its domains, namely PDZ and DEP. When the Wnt molecule binds to the frizzled domain, it starts a cascade that exposes Dishevelled’s DIX domain and creates a site that can completely hook Axin. The oligomer of axin is titrated away, and the β-catenin complex is destroyed by Dsh [143].

Frizzled has essential cytoplasmatic segments containing a GSK-3 pseudo-substrate sequence (Pro-Pro-Pro-Ser-Pro-x-Ser) that is pre-phosphorylated by CKI, namely LRP5 and LRP6 proteins [144]. Once bound to the receptor complex, Axin binds to β-catenin and activates GSK-3. In this way, the Axin-bound receptor prevents β-catenin-mediated phosphorylation. Therefore, the β-catenin is damaged yet continues to be produced up to an increased concentration. Once the level of β-catenin rises sufficiently to saturate all sites in the cytoplasm, β-catenin is translocated into the cell nucleus. After involving the transcription factors of LEF1, TCF1, TCF2, or TCF3, β-catenin releases other transcription factors, binds to transcriptional activators, and then switches to target genes [144].

#### 6.1.2. Canonical Signaling Pathway

Canonical Wnt signal activates when the Wnt ligand binds to the Fz receptor and the LRP coreceptor, activating Dvl and inhibiting GSK-3β. Without a Wnt ligand, the activation of Wnt signaling does not occur; GSK-3β, axin, and APC bind β-catenin. This binding stimulates the phosphorylation of β-catenin by GSK-3β and is followed by the ubiquitinylation of β-catenin, which is further degraded by proteasomes. β-catenin accumulates in the nucleus and binds to the T cell factor and lymphoid enhancer factors (TCF/LEF) to regulate gene transcription [145]. Figure 3A,B illustrate the canonical signaling pathway.

### 6.2. PGE2 Elevates the Accumulation of β-Catenin in Lung Injury

PGE2 levels in the lung compared with other organs tend to be higher, as evidenced by the isolation of alveolar epithelial cells that release high amounts of PGE2 [146]. According to in vitro studies using cell lines performed by Fujino et al. [126] and Fang et al. [147], PGE2 was shown to stimulate a double signaling cascade, comprising the activation of PI3K and protein kinase Akt by the free G-protein βγ subunit and the direct binding of the G-protein αs subunit to Axin. Although the activity of GSK-3β is regulated by these two pathways independently, the activation of GSK-3β reduces β-catenin signaling by promoting the phosphorylation of β-catenin, which is subsequently degraded by the 26S proteasome. Akt phosphorylation via PI3K signaling inhibits the activation of GSK-3β and APC in the protein subunits of the G-protein-linked PGE2 receptor EP2, as well as associating the activated α subunit of Gs with the RGS domain of Axin, so that the axin–GSK–3β complex releases β-catenin [122,126,147]. When PGE2 is activated in these two pathways, unphosphorylated β-catenin accumulates in the cytoplasm. T cell factor-4 (TCF4) and hypoxia-inducible factor-1 (HIF1) are two genes that β-catenin activates once it enters the nucleus, and this promotes cell survival, proliferation, and angiogenesis [122,126,127,128]. An in vitro study using human colon cancer cells conducted by Shao et al. [148] demonstrated that PGE2 activation escalates GSK-3 phosphorylation.

Consequently, β-catenin is accumulated, and the expression of T cell-4 transcription factor is induced, which forms a transcriptionally active complex with β-catenin. Furthermore, similar to the Wnt/β-catenin pathway, the translocation of β-catenin into the cell nucleus releases transcription factors, which then bind to transcriptional activators and switch to target genes [144]. Figure 3C illustrates the mechanisms by which PGE2 promotes immunoregulation through the activation of Wnt-β-catenin signaling.

### 6.3. Cross-Regulation between β-Catenin and NF-κB Signaling Pathways

The cross-regulation between the Wnt/β-catenin and NF-κB signaling pathways is essential to the regulation of diverse genes and active pathways, e.g., many physiological and pathological effects, regulations associated with the areas of growth, immune function, inflammation, tumorigenesis, tumor invasion, and metastasis, as well as cardiovascular and bone disease. However, the activity and signaling consequences are also regulated by direct interactions between the two pathways, resulting in diversity and complexity [130]. β-catenin and NF-κB can activate the expression of the gene-inducible nitric oxide synthase (iNOS), yet β-catenin also has an inhibiting effect on NF-κB. The functional cross-regulatory mechanisms between these two pathways play a complex role in Wnt/β catenin and NF-κB signaling in the pathogenesis of certain cancers and other diseases [130]. β-catenin plays an essential role in the development and function of homeostasis. Deng et al. [131] showed that catenin application in colon cancer decreased the transactivation activity and the expression of NF-κB target genes and inhibited the expression of NF-κB target genes, including Fas and TRAF1. 

A colonic epithelial cell infection study with Salmonella typhimurium demonstrated the activity of β-catenin, which is able to inhibit NF-κB via two possible mechanisms: by stabilizing IκBα or by binding to NF-κB subunit p50, which prevents its transport from the cytoplasm to the nucleus and also occurs in the nucleus. Both mechanisms can cause a decrease in the transcriptional activity of NF-κB. The inhibition of NF-κB by β-catenin activation reduces inflammation. The study found that the infection of colonic epithelial cells with Salmonella typhimurium caused an elevation in GSK-3β-dependent β-catenin phosphorylation, leading to the up-regulation of Wnt2, IL-6, and IL-8 via TLR5, and thus activating NF-κB [132]. Liu et al. [133] proved that the ectopic over-expression of Wnt11 and Wnt2 decreased the transcriptional activity of NF-κB and AP1. This evidence suggests that Wnt 11 and Wnt2 activity triggers a negative feedback mechanism that controls NF-κB through β-catenin activation. Figure 3D illustrates the cross-regulation between the β-catenin and NF-κB signaling pathways.

## 7. The Mechanisms of Immunoregulation after Exogenous MSCs Administration on Virus-Induced Acute Lung Injury 

Based on the available literature, the administration of MSCs could enhance endogenous PGE2, escalate the number of cells expressing β-catenin, and decrease the NF-κB transcription factor and TNFα pro-inflammatory cytokine expression, as well as IL-1β. 

As previously explained, the characteristics of the innate immune response are the primary determinant of the outcome of influenza infection. PAMPs of the influenza A virus are recognized by three main classes of PRRs, including the cytosolic sensor RIG-I [34], TLR7 in endosomes [35], and NLR molecules [36]. These three pathways can induce the production of pro-inflammatory cytokines via NF-κB activation, which causes damage to lung pathology [33,36]. The inhibition of the NF-κB transcription factor inhibits the production of excessive pro-inflammatory cytokines due to exposure to the highly pathogenic influenza A virus, thereby preventing further tissue damage. MSCs exposure to pro-inflammatory cytokines, such as IFγ and TNfα in MSCs cultures, can increase the expression of PGE2 and other molecules [89]. According to the previously mentioned literature, the repair mechanisms of acute lung damage induced by the highly pathogenic influenza A virus begin with the release of excessive pro-inflammatory cytokines (cytokines storm), which stimulates allotransplanted MSCs to produce large amounts of active biomolecules. One of the active biomolecules is transforming growth factor-β (TGF-β), which can increase endogenous PGE2 expression by inducing COX2. A study by Fang et al. [149] suggested that TGF-β regulates COX-2 expression and PGE2 production in human granulosa cells.

The production of active biomolecules of PGE2 increases the accumulation of β-catenin in the cytoplasm. As previously mentioned, PGE2 has been proven to stimulate a double signaling cascade involving the activation of PI3K and protein kinase Akt by the free G-protein βγ subunit and the direct association of the G-protein αs subunit to Axin. The activation of GSK-3β promotes the phosphorylation of β-catenin and subsequent degradation by the 26S proteasome, reducing β-catenin signaling. On the other hand, Akt phosphorylation via PI3K signaling inhibits the activation of GSK-3β and APC in the protein subunits of G-protein-linked PGE2 receptor EP2, as well as the association of the activated α subunit of Gs with the RGS domain of Axin, so that the Axin–GSK–3β complex releases β-catenin [122,126,147].

In the further course of the disease, other pathways are needed to increase the activation of β-catenin, which serves to maintain homeostatic function in severe lung injury. The increase in β-catenin through the trans-differentiation process of the canonical Wnt/β-catenin pathway, which is usually involved in cell development through frizzled receptors, has a similar effect on PGE2, which functions to decrease the arrangement of the Axin-GSK-3β-APC complex, thereby stabilizing the accumulation of β-catenin in the cytoplasm [126,128,147]. The trans-differentiation process is characterized by the increased expressions of several Wnt ligands, the increased phosphorylation of Wnt signaling intermediates (i.e., LRP6 and DVL3), increased transcriptionally active β-catenin, and the increased expression of β-catenin-driven target genes that may initially be beneficial in lung injury [150,151].

The β-catenin protein plays a vital role in the process of immunoregulation. The inhibition of NF-κB transcription factor activity is a β-catenin pathway that plays a role in the immunoregulation mechanism. According to previous studies [84,131,148], β-catenin plays a vital role in homeostatic function by inhibiting NF-κB through cross-regulatory mechanisms. This cross-regulation is essential for regulating various genes and their active pathway, which involves controlling many physiological and pathological effects related to immune and inflammation functions. 

Decreased NF-κB activity also has a significant direct effect on RAGE expression. The activation of the NF-κB pathway through RAGE can provide positive feedback. Furthermore, NF-κB can regulate the activation of RAGE, which causes an elevation of RAGE expression. This positive feedback cycle can further increase inflammation and tissue injury. These conditions occur mainly in diseases with elevated levels of the RAGE ligand [152,153]. Other studies have stated that HMGB1 signaling results in NF-κB expression. This activation promotes inflammation through a positive feedback loop because NF-κB can elevate the activation of various receptors, including RAGE [154,155]. Figure 4 illustrates the mechanisms of immunoregulation after exogenous MSCs administration on virus-induced acute lung injury.

## 8. Conclusions

The characteristics of innate immune response are essential to tissue damage caused by highly pathogenic influenza A virus infection. ARDS involves damage in the alveolar–epithelial–endothelial barrier of the lung, which causes the leakage of proteins of intravascular origin into the alveolar lumen. In addition to PAMPs, DAMPs are host molecules that regulate the activation of PRRs during influenza A virus infection. Activation of this molecule, in turn, triggers a signaling cascade that promotes the transcription factor NF-κB, which encodes pro-inflammatory cytokines and chemokines. Allotransplanted MSCs may exert a beneficial effect on tissue injury by inhibiting the transcription factor NF-κB. The effect of allotransplanted MSCs against inhibitory transcription factor NF-κB is known to operate through several related mechanisms, via the release of PGE2 and the accumulation of β-catenin.

The β-catenin protein plays a crucial role in the process of immunoregulation. Inhibition of the activity of transcription factor NF-κB is one of the β-catenin pathways that play a role in the mechanism of immunoregulation. The cross-regulatory mechanism of β-catenin in the cytoplasm activates Iκβ, thereby inhibiting NF-κB activity. However, there are still many other pathways that are involved in the mechanism of immunoregulation through inhibiting the innate immune system in addition to inhibiting NF-κB activation. Thus, the utilization of these approaches in acute lung damage caused by the highly pathogenic influenza A virus requires further research both in animal models and humans. Furthermore, the innate immune system is likely to activate its adaptive function to perpetuate the process. Therefore, it is necessary to evaluate the effects of MSCs on the adaptive immune system. Hopefully, a better understanding can be formed of the mechanisms of pro-inflammatory pathways that contribute to lung injury and host defenses, which can inhibit inflammatory damage while maintaining the host defense system. The mechanism of immunoregulation through β-catenin activation requires further research into the development of methods that can modify MSCs in vitro before transplantation, one of which involves genetic modification through the transfection method, such that MSCs can be made to carry the β-catenin gene. Another alternative approach is to provide combination therapy with β-catenin and MSCs.

## Figures and Tables

**Figure 1 ijms-24-07299-f001:**
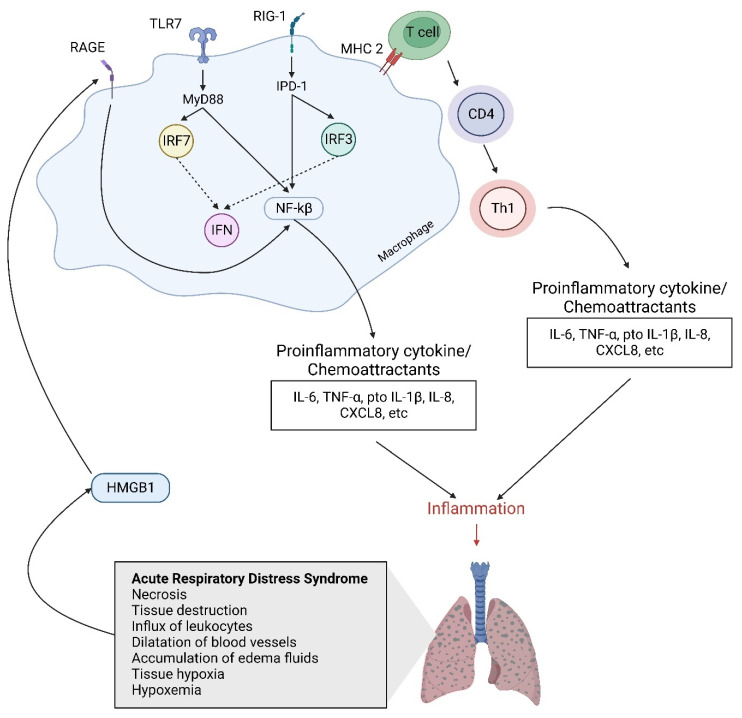
Cytokine Storm and ALI/ARDS Induced by Influenza Virus Infection. When a highly pathogenic influenza A virus infects alveolar epithelial cells, macrophages recognize this virus using TLR 7, which involves several adaptor molecules and transcription factors, such as MyD88, and activates IRF7 and NF-κB. Influenza A virus can also be recognized by RIG 1 via the adaptor molecule IPD-I and activate IRF3 and NF-κB. The activation of IRF-7 and IRF-3 causes their translocation into the nucleus. This produces interferon type I, while NF-κB acts as a transcription factor to induce pro-inflammatory cytokines’ production, including IL-6, TNFα, and pro-IL-1β. Macrophages also present virus peptides on the cell surface through MHC class II and activate lymphocyte CD4 T cells, which differentiate into Th1. Activated lymphocyte T cells then release various pro-inflammatory cytokines and chemoattractants. All of these mechanisms result in a cytokine storm. The cytokine storm then leads to ARDS, characterized by necrosis, tissue damage, an influx of leucocytes, and blood vessel dilation, causing edema fluids to accumulate in the alveolar, characterized by tissue hypoxia and hypoxemia. The damaged tissue releases HMGB1 and binds with RAGE, activating the intracellular signaling pathway and resulting in NF-κB activation and pro-inflammatory gene expression induction; thus, the cytokine storm continues. We created this figure using the BioRender online app and license.

**Figure 2 ijms-24-07299-f002:**
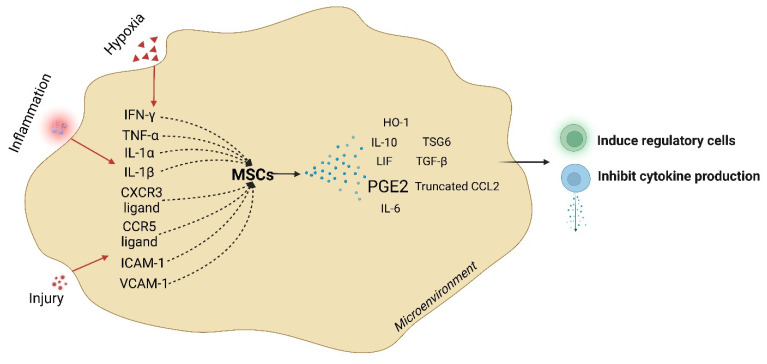
Communication of MSCs in Damaged Tissue and the Immunosuppression Property of MSCs. Tissue injury, inflammation, and hypoxia factors cue a signal that mobilizes MSCs to the damaged tissue. Once MSCs enter the microenvironment of the injured tissue, IFNγ, together with pro-inflammatory cytokines, such as TNFα, IL-1α, and IL-1β, the occurrence of a burst of chemokine, and the expression of adhesion molecules, stimulate MSCs to release large amounts of immunosuppressive factors. These immunosuppressive factors then promote the activation of immune cells, which induce regulatory cells and inhibit cytokine production. This figure was created using the BioRender online app and license.

**Figure 3 ijms-24-07299-f003:**
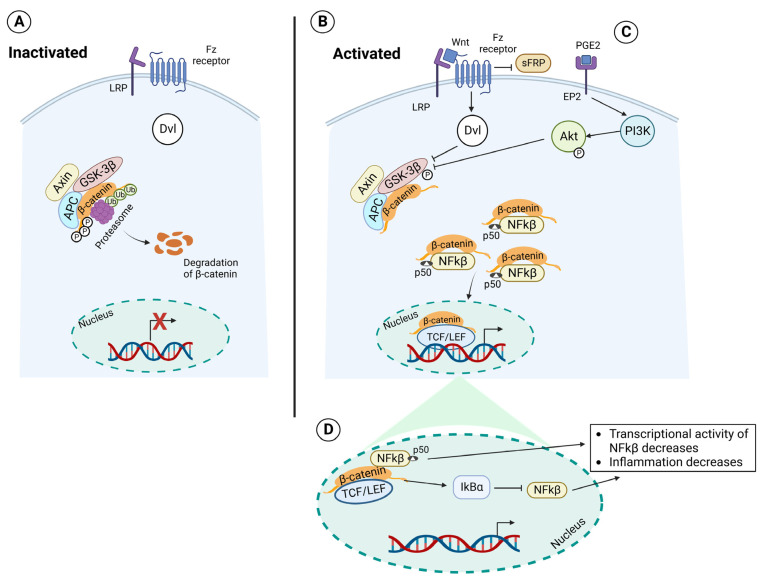
PGE2 Promotes Immunoregulation through Cross-Regulation between β-catenin and NF-κB by Activating the Wnt-β-catenin Signaling Pathways. (**A**) Without the Wnt ligand, β-catenin is bound by GSK-3β, Axin, and APC. This binding-stimulated phosphorylation of β-catenin by GSK-3β leads to the ubiquitination of β-catenin, which is further degraded by proteasomes. (**B**) When the Wnt ligand binds to the Fz receptor and the LRP co-receptor, activation of the canonical Wnt signal occurs, followed by activation of Dvl and inhibition of GSK-3β. To regulate gene transcription, β-catenin accumulates in the nucleus and binds to T cell factor and lymphoid enhancer factors (TCF/LEF). (**C**) PGE2 stimulates a double signaling cascade through the EP2 receptor via PI3K and protein kinase Akt activation involving the free G-protein βγ subunit and the direct association of the G-protein αs subunit to Axin. Following PGE2 activation in these two pathways, unphosphorylated β-catenin accumulates in the cytoplasm, then enters the nucleus and inhibits the transcription factor of NF-κB (**D**). The activity of β-catenin can inhibit NF-κB via two possible mechanisms—by stabilizing IκBα or by binding to NF-κB subunit p50—which prevents its transport from the cytoplasm to the nucleus and also occurs in the nucleus. Both mechanisms can cause a decrease in the transcriptional activity of NF-κB. The inhibition of NF-κB by β-catenin activation reduces inflammation. This figure was created using the BioRender online app and license.

**Figure 4 ijms-24-07299-f004:**
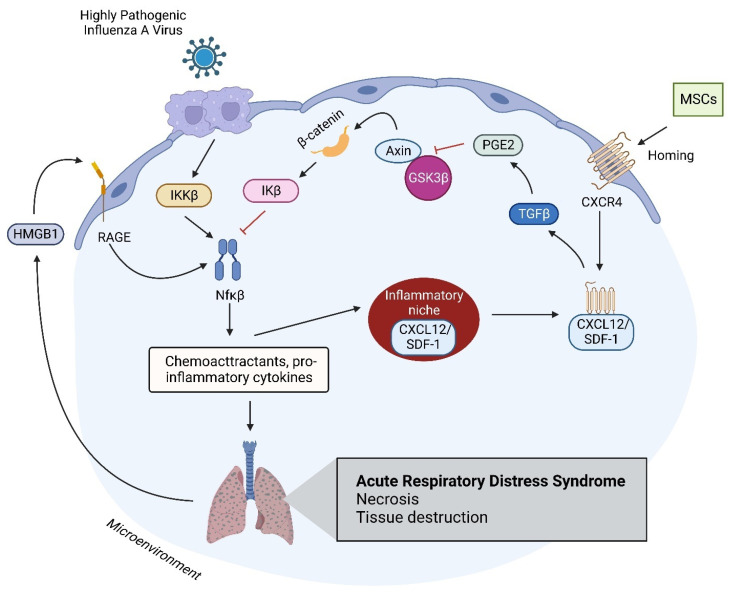
Immunoregulation Mechanisms of Exogenous MSCs on Virus-Induced Acute Lung Injury. A single-stranded RNA of the highly pathogenic influenza A virus binding with the host initiates several signaling transductions. IKKβ is then activated, which can lead to the phosphorylation and degradation of Iκβ, causing the active transcription factor NF-κB to release. NF-κB encodes pro-inflammatory cytokines and chemokine attractants, which lead to ARDS. The injured cells release HMGB1, which has a high affinity to RAGE ligands. HMGB1 bonds with RAGE, and this cues a signal to activate NF-κB, which releases adhesion molecules and pro-inflammatory cytokines. CXCL12 (SDF-1) regulation increases in the inflammation niche due to pro-inflammatory cytokine stimulation. CXCL12 (SDF-1) interacts with its ligand CXCR4, which is expressed by exogenous MSCs and starts the homing process. Active biomolecules of endogenous PGE2 are then produced in large amounts, which leads to the phosphorylation of GSK-3 such that β-catenin accumulates in the cytoplasm. These accumulations inhibit NF-κB via the activation of Iκβ, and as a consequence, the release of pro-inflammatory cytokines and chemokines is inhibited. This figure was created using the BioRender online app and license.

**Table 1 ijms-24-07299-t001:** The study of MSCs in lung injury induced by different sub-types of influenza virus.

No	Author	Virus	MSCS Sources	Outcome of Infection	Study Design	References
1	Chen, J., et al. (2020)	Influenza A (H7N9)	Allogeneic, menstrual-blood-derived MSCs	Lower levels of PCT, serum creatinin (sCr), creatine kinase (CK), prothrombin time (PT), and D-dimer; hemoglobin (Hb) up-regulated; improvement on CCT	Human	[110]
2	Yudhawati, R., et al. (2020)	Avian influenza A/H5N1 of A/turkey/East Java/Av154/2013	Bone-marrow-derived mesenchymal stem cells (BM-MSCs)	Lower levels of lung alveolar protein, PaO2/FiO2 ratio, and histopathological score; suppressed expressions of NF-κB, RAGE, TNFα, and IL-1β; Sftpc and Aqp5+ enhanced	In vivo (mice)	[111]
3	Bogatcheva and Coleman (2021)	Influenza A Virus H1N1	Adipose stromal cells	Limits pulmonary histopathological changes, infiltration of inflammatory cells, protein leak, water accumulation, and arterial oxygen saturation (SpO2) reduction; significant suppression of IL-6 and MCP-1; viral antigen, BAL TNF-α, PDL1, and Angpt2 levels lowered	In vivo (mice)	[112]
4	Chan et al. (2016)	H5N1 influenza A viruses A/Hong Kong/483/97 (483/97), A/Hong Kong/486/97 (486/97), and A/Vietnam/3046/04 (3046/04)	Bone-marrow-derived human MSCs	Transporter proteins CFTR and α1Na,K-ATPase enhanced; Ang1 in MSCS supernatants, KGF secretion, CFTR protein expression increased; lower CD4+ T cells, NK cells, IP10, MCP-1, MCP-3, M1P-1α, RANTES, IL-4, IL-17, and TNF-α; more macrophages/monocytes	In vitro and in vivo	[27]
5	Khatri et al. (2018)	Influenza virus H1N1, H3N2, H9N5	Isolated extracellular vesicles EVs from swine bone-marrow-derived MSCs	Reduced influenza virus replication; lower levels of TNF-α and CXCL10; cytokine IL-10 were higher	In vivo (pigs)	[113]
6	Li, Y., et al. (2016)	A/Hong Kong/2108/2003 [H9N2 (HK)] H9N2 virus	Adipocyte, chondrocytes, and osteocytes murine MSCs	Higher PaO2, SaO2 and pH values; PaCO2, GM-CSF, MCP-1, KC, MIP-1α, MIG, IL-1α, IL-6, IL-10, TNF-α, IFN-γ, ERK, JNK, CD14, and TLR4 expression were lowered	In vivo (mice)	[114]
7	Loy, H., et al. (2019)	Influenza A/Hong Kong/483/97(H5N1)	Human umbilical cord MSCs	In vitro: reduced IL-β, IFN-λ1, IL-6, IL-1β, IFN-γ-induced protein 10 (IP-10), MCP-1, and RANTES; enhances responses of IL-4, IL-10, IL-11, IL-13, IL-RA, IL-10, IL-11, and IL-RA. In vivo: greatest significance for IP-10, MCP-1, RANTES, and IL-6; lower concentrations of Evans blue dye, IP-10, RANTES, TNF-α, MCP-1, IL-1β, IL-6, and IL-8	In vitro and in vivo (mice)	[115]

## Data Availability

Not applicable.

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
