# Peer review of "PGE2 Produced by Exogenous MSCs Promotes Immunoregulation in ARDS Induced by Highly Pathogenic Influenza A through Activation of the Wnt-β-Catenin Signaling Pathway"

_ijms, 2023, doi:10.3390/ijms24087299_

Round 1
Reviewer 1 Report
see attached comments

Author Response
Editor
International Journal of Molecular Sciences
03 April 2023
Subject: Revision and Resubmission of manuscript ijms-2233561
Dear Editor,
Thank you for the opportunity to revise our paper "PGE Promotes Immunoregulation of Exogenous Mesenchymal Stem Cells Through Activation of Wnt-β-catenin Signaling in Acute Respiratory Distress Syndrome Induced by Highly Pathogenic Influenza A Virus”. We are most grateful to you and the reviewers for the constructive comments and have made amendments accordingly.
I have included the editor comments immediately after this letter and responded to each of the points, indicating how we have addressed concerns and describing the changes we have made.
It is our sincere hope that the revised manuscript is now suitable for publication in International Journal of Molecular Sciences. We thank you for your continued interest in our research and look forward to hearing from you very soon.
Respectfully Yours,
Dr. Resti Yudhawati
Department of Pulmonology and Respiratory Medicine, Faculty of Medicine, Universitas Airlangga – Dr Soetomo General Academic Hospital
Jl. Prof. Dr Moestopo 6-8 Surabaya, 60286, Indonesia
Tel: +6289673691726
E-mail: resti-y-m@fk.unair.ac.id
Reviewers Comments:
Thank you for addressing the comment.
Reviewer 1 comments:
- Title is very long, not clear. Simplify it such as “PGE2 from Treatment MSCs Promotes Immunoregulation of Acute ARDS Induced by Influenza A Through Activation of the Wnt-βcatenin Signaling Pathway.
Response: After long consideration, we decided to revise the title to “PGE2 Promotes Exogenous MSCs Immunoregulation of Acute ARDS Induced by Highly Pathogenic Influenza A Through Activation of the Wnt-β-catenin Signaling Pathway”
- Page 1
Abstract add all key words in abstract for search ability. Include NFkb
Response: The amendments has been made accordingly in Abstract section.
End of 4th sentence - add (DAMPs activation).
Response: The 4th sentences refers to Nfκβ activation. Following the comment, we realize that the sentence is a bit confusing, so we added “Nfκβ activation” at the end of 4th sentences.
- Page 2
First full paragraph-confusing, please rewrite such as Developing treatments prioritizing immunomodulation and inhibiting the intracellular signaling cascade, as well as limiting viral replication is essential because the patient infected with highly pathogenic influenza virus already has severe complications at admission. (Combine with second paragraph Cell-based therapy…. Allotransplantation UTILIZING MSC has shown beneficial effects in ARDS [11]. Furthermore, MScs have low immune….
Also Researches on MSCs administration… or
Response: Following the suggestion, the amendments has been made accordingly and can be checked in line 51–60.
- Last 2 sentences in this paragraph – …H5N1 virus proves indicated that MSCs…However the mechanism that leads to MSC mobilization to damaged tissues remains unclear.
Response: Following the suggestion, the amendments has been made and can be checked in line 71–75.
- Host Immune Response…
2nd line an enveloped corona virus with a diameter…
Response: Following the suggestion, the amendment has been made accordingly. (Line 91)
Last sentence of paragraph - On the other hand, several numbers of transcripts including PB2, PB1, and PA are coded by the viral RNA genome, which are essential for the replica…
Response: After long consideration, we decided to remove this sentence.
The characteristic of innate immune response is the primary determinant of the outcome of influenza infection.
Response: Following the suggestion, the amendment has been made accordingly. (Line 104)
These three pathways can induce pro-inflammatory cytokines and chemokines, production which leads to pathological lung damage.
Response: Following the suggestion, the amendment has been made accordingly. (Line 110–112)
HA protein can directly express induce expression of several host cytokines, including… Host receptors can recognize…
Response: Following the suggestion, the amendment has been made accordingly. (Line 114–115)
- Page 3
1st paragraph - PKR has roles in the apoptosis process of influenza A virus infected cells…
Response: Following the suggestion, the amendment has been made accordingly. (Line 119–120)
Line 7 NFkB subunit p65/p50 can induce pro-inflammatory cytokines, chemokines, and proapoptotic adhesion molecule INFb [31]. (signaling, not an adhesion molecule)
Response: Following the suggestion, the amendment has been made accordingly. (Line 124–126)
3rd paragraph last sentence – These antigen presentations and recognition by T cells activate the adaptive immune response.
Response: Following the suggestion, the amendment has been made accordingly. (Line 145–147)
- Cytokine Storm
1st sentence …virus infection cause are causing pathological lung damage.
Response: Following the suggestion, the amendment has been made accordingly. (Line 158)
2nd paragraph – “However, this is…tissue” is unclear better to say this cytokine storm causes local cell death and severe damage to lung tissue and eventually results in fibrosis.
Response: Following the suggestion, the amendment has been made accordingly. (Line 160–162)
A few sentences later-hyalin membrane formation and fibrosis.Or fibroblast accumulation not formation
Response: Following the suggestion, the amendment has been made accordingly. (Line 165–166)
Last paragraph - The mechanisms of cytokine storm damage goes through multiple stages.
Note - the next two paragraphs on page 4 are unnecessary and difficult to read - rewrite or eliminate.
Response: Following the suggestion, the amendment has been made and can be checked in line 170–171. As for the next two paragraphs, we decided to rewrite it instead of eliminate because the role of DAMPs is crucial in the mechanism of ARDS occurrence, since it gives positive feedback to the activation of Nf-κB which then will be inhibited by MSCs.
- Page 6
- Stem Cells for Lung Therapy
Stem cells are undifferentiated cells with several differentiation potentials.
2nd paragraph Chemokines are small peptide molecules that intitiates……….owned expressed by the stem cell [42].
4th paragraph Evidence based research has shown that embryonic and mature cells can be induced to differentiate…
These variables include the route of administration, the MSCs origin source and… MSCs are introduced in section 5 so this paragraph is still general to all stem cells.
LAST SENTENCE Currently, stem cells extensively tested for ARDS include hematopoietic stem cells (HSCs) and mesenchymal stem cells (MSCs)[46].
Response: After long consideration, we decided to remove this section.
- Page 7
…mesodermal tissues.
Response: Following the suggestion, the amendment has been made accordingly. (Line 298)
…from bone marrow, umbilical cord (blood or Whartons jelly) and fatty tissues.
Response: Following the suggestion, the amendment has been made accordingly. (Line 301)
2nd paragraph
MSCs, sometimes also referred to as marrow stromal cells, were first identified in the guinea pig by Alexander Friedenstein in 1968 [50]. The first human MSCs were isolated by Haynesworth and Caplan in 1991 [REF].
Response: Following the suggestion, the amendment has been made accordingly. (Line 308–310)
…following criteria: 1) must grow be plasticity while attached 2) must express specific cell surface markers that include CD73, CD90 and CD105, but not express the hematopoietic markers CD11b, CD14, CD34, or CD45 3) must have the capacity to differentiate to mesenchymal lineages including adipocytes, osteoblasts and chondrocytes under defined in vitro conditions. These characteristics of isolated human MSCs were all originally described by Pittenger et al (REF - 1999 Science 284:143-147).
(NOTE- these criteria distinguish MSCs from known HSC characteristics. HSCs do not attach and grow, do not express these surface markers, or differentiate to mesenchymal lineages under these conditions)
Response: Following the suggestion, the amendment has been made accordingly. (Line 313–320)
1 Allogeneic MSCs
Allogeneic MSCs are exogenous stem cells donors from the same species but are not…..
(NOTE to authors – While treatment with allo-MSCs does have certain advantages, allo or auto MSCs, it is still difficult to ensure engraftment in the host and most grafted cells die in the first 72 hrs thus limiting effective treatment.)
Response: Following the suggestion, the amendment has been made accordingly (Line 329–330). As for the ability of engraftment of MSCs is explained in section 4.3. Several studies mentioned that MSCs only have engraftment ability less than 5%, however the significance role by MSCs related to immunoregulation is “the paracrine effect” which the mechanisms are explained in more detail in this review article.
- Page 7 Paragraph 5
The first study to mix MSCs with isolated immune cells – T, B, NK- and show that each responded to MSCs was 2005 Blood 105:1815-1822, and this was also the first identification of PGE2 as effector released from MSCs.
Response: Following the suggestion, the amendment has been made accordingly. (Line 342–346)
- Page 8
MSCs do express MHC-II RNAs but they are not translated to proteins. Under INFb treatment MSCs will express MHC-II proteins and WILL stimulate T cells although the response is blunted by the other immunomodulatory molecules expressed by MSCs.
Response: Following the comment, we revised and added the explanation of this mechanisms in line 367–385.
Line 3617–385 in revised manuscript:
MSCs exhibit low levels of MHC class I molecules on their surface, and lack expressions of MHC class II as well as the co-stimulatory molecules CD80 (B7-1), CD86 (B7-2), and CD40, which may limit immune recognition. However, under infection conditions, during which pro-inflammatory factors are released and inflammation occurs, the expression of MHC-II can be up-regulated in antigen-presenting cells (APCs) [62]. Klyushnenkova et al. supported this statement after demonstrating that MSCs treated with IFN-γ showed up-regulated MHC-II molecules [63]. Additionally, MSCs under inflammatory circumstances show an up-regulation of the co-stimulatory molecule CD40. However, T cells are not fully activated because CD40 expression is counterbalanced by the inhibitory molecule CD274 (also known as PD-L1), which is also expressed by MSCs [64]. The hypothesis that allo-MSCs have the same efficacy as auto-MSCs has been proven. However, in vivo studies have revealed that allo-MSCs are not completely immunogenic, thus possibly eliciting an immune response despite their low immunosuppressive and immunogenicity. In vivo and in vitro studies have both reported these findings. However, until recently, this has been a point of contention, with inconclusive results [65,66].

Reviewer 2 Report
The objective of the review article is very important to the scientific community. The authors have started their manuscript broadly introducing MSC, covering basic aspects until they reached the point of discussion. The general part is slightly longer than one should expect from the focused title (almost half the manuscript) and not adding any new. It is recommended to be more focused on MSC aspects related to the point of interest.
The authors had missed crucial points regarding MSC infusion which is the mode of MSC infusion. Indeed mode of delivery and homing of MSC influence their biological activity. There are more than one mode of MSC delivery upon which MSC reaction/secretion would differ.
Beta catenin signaling is a point of great interest in MSC homing together with PGE2 priming of MSC, points which are not discussed by the authors.
The article is missing the explanation of the double sword effect of those cells which can be stimulators and also regenerative in certain conditions depending on the stimulus received by the cell from the homing environment. MSC side effects is lacking throughout the manuscript.
More translational points should be added to the review article in order to enhance the clinical aspect of those cells and advances in the therapeutic protocols. Cell count/Ratio of infusion and timings in relation to the disease course is related to the amounts of mediators released and hence the amount of PGE2 (concentration) that would influence an inhibitory rather than stimulatory effects.
More recent references and human clinical trials are missing from the references. Only one human clinical trial is mentioned though the literature has many others.
English language needs to be revised
Author Response
Editor
International Journal of Molecular Sciences
03 April 2023
Subject: Revision and Resubmission of manuscript ijms-2233561
Dear Editor,
Thank you for the opportunity to revise our paper "PGE Promotes Immunoregulation of Exogenous Mesenchymal Stem Cells Through Activation of Wnt-β-catenin Signaling in Acute Respiratory Distress Syndrome Induced by Highly Pathogenic Influenza A Virus”. We are most grateful to you and the reviewers for the constructive comments and have made amendments accordingly.
I have included the editor comments immediately after this letter and responded to each of the points, indicating how we have addressed concerns and describing the changes we have made.
It is our sincere hope that the revised manuscript is now suitable for publication in International Journal of Molecular Sciences. We thank you for your continued interest in our research and look forward to hearing from you very soon.
Respectfully Yours,
Dr. Resti Yudhawati
Department of Pulmonology and Respiratory Medicine, Faculty of Medicine, Universitas Airlangga – Dr Soetomo General Academic Hospital
Jl. Prof. Dr Moestopo 6-8 Surabaya, 60286, Indonesia
Tel: +6289673691726
E-mail: resti-y-m@fk.unair.ac.id
Reviewers Comments:
Thank you for addressing the comment.
Reviewer 2 comments:
- The objective of the review article is very important to the scientific community. The authors have started their manuscript broadly introducing MSC, covering basic aspects until they reached the point of discussion. The general part is slightly longer than one should expect from the focused title (almost half the manuscript and not adding any new. It is recommended to be more focused on MSC aspects related to the point of interest.
Response: The point of this review article is the role of stem cells against ARDS, hence we believe that it is important to describe the mechanisms of ARDS occurrence in brief yet detail to the readers. So, hopefully the mechanisms of MSC as an immunoregulatory will be discussed completely in this review article. The section of “Stem Cells” is eliminated and replaced by sub-topic of Mesenchymal Stem Cells with more detailed explanation followed by the explanation of mechanisms of immunoregulatory by MSC related to the mechanisms of cytokine storm.
- The authors had missed crucial points regarding MSC infusion which is the mode of MSC infusion. Indeed mode of delivery and homing of MSC influence their biological activity. There are more than one mode of MSC delivery upon which MSC reaction/secretion would differ.
Response: Following the suggestion, we added the explanation of MSCs Mode and Delivery in section 4.2 (line 386–431).
Line 386–431 in revised manuscript:
4.2. MSC Mode and Delivery
There are several techniques for MSCs administration, depending on the underlying clinical indications and pathological abnormalities. The proper selection of the mode of delivering MSCs provides optimal therapeutic efficacy [67]. There are two main methods used to deliver cells into the body: local delivery into the tissue, and systemic delivery. Local delivery is via injection, for example intraperitoneal (IP), intramuscular (IM), or intracardiac, while systemic delivery methods involve the administration of cells intravenously (IV) or via intraarterial (IA) paths [68]. The alternative route of delivery is direct injection (DI) into the tissue or organ targeted, in order to escalate the engraftment and/or local differentiation [69].
Another method commonly used to deliver MSCs is via intravascular (IV) infusion, owing to its ease of utilization and low risk [70]. However, cells administered through IV have to pass through the lungs first before being distributed to the rest of the body. This condition leads to a substantial problem called the pulmonary “first-pass” effect, which produces high numbers of entrapment cells [71]. This situation occurs because MSCs are approximately 20-30 µm in diameter, and most particles of this size are filtered out by the lungs [71]. Nevertheless, MSCs entrapment in the lungs provides therapeutic effects on the targeted tissues. A concise review study concluded that MSCs isolated from bone marrow alter the tissue microenvironment by secreting soluble factors (autocrine-paracrine). This makes a major contribution to tissue improvement via the trans-differentiation capacity [72]. Several studies have proven that engraftment plays an important role in MSCs therapy aimed at injury; however, this engraftment ability may not play a key therapeutic role in the lungs. Because lungs’ engraftment ability is less than 5%, the paracrine mechanisms play greater roles [73,74]. The beneficial effects of MSCs are derived from their capacity to secrete paracrine factors that modulate immune response and alter the response of the epi-thelium, endothelium, and inflammation cells to injury. Administration via IV has been extensively re-ported to confer therapeutic effects through mechanisms involving secondary signaling effector cell systems and interactions with the host immune systems [69].
Theoretically, the intraarterial (IA) delivery route enables cells to pass through the lungs at least once, thus avoiding the pulmonary “first-pass” effect, and consequently more cells are able to reach the targeted tissues [68].
Several studies have reported the positive therapeutic effects of MSCs; however, there are still a lot of questions and challenges related to the application of MSCs. Cell therapy can induce cell emboli, which may potentially increase mortality. This type of embolism depends on the concentrations and rates of cell administration [69,75]. Other risks may arise, such as MSCs transformation, potential adverse inflammatory effects, tumor formation, and thrombosis associated with the intravenous infusion of MSCs [70]. Until recently, the safety of MSCs therapy has been under question, but the efficacy and the resulting interaction in the microenvironment of the host remain controversial.
- Beta catenin signaling Is a point of great interest in MSC homing together with PGE2 priming of MSC, points which are not discussed by the authors.
Response: Following the comment, we added the explanation of beta catenin signaling in section 6, line 825–860. Regarding PGE2 priming will be explained in more detail in section 6.2 and 6.3.
Line 825–860 in revised manuscript:
The ?-catenin protein is a subunit of the cadherin protein complex that acts as an intracellular signal transducer, which is also involved in the regeneration and immunoregulation process. ?-catenin is a double-function protein that regulates the coordination of cell adhesion and the transcription of genes. ?-catenin is homologue with γ-catenin also known as plakgoblin, and is expressed in various tissues [137]. The Wnt/?-catenin signaling pathway is a crucial pathway for several biological processes. The aberration of this signaling can be found in several human diseases [138].
Wnt/ ?-catenin signaling also acts as the regulation pathway of the stem cells niche in several tissues. However, the role of Wnt signaling in the immunoregulatory function has not been extensively reported, although it is broadly involved in the regulation of adult MSCs’s fate in various tissues. Through the canonical pathway, this Wnt/ ?-catenin pathway helps the maintenance of MSCs, and improves their proliferation ability as a function of regeneration [29].
The available studies suggest that one of the main roles of Wnt signaling is determining the fate of and controlling stem cells [139]. Gehart and Clevers [140] reported one of the main communicative roles in the stem cells niche is played by the Wnt/ ?-catenin pathway. The activation of the canonical Wnt pathway, which is dependant on ?-catenin, is essential to the biology of MSCs, and induces differentiation and proliferation [139].
The elevation of ?-catenin under certain tissue damage conditions is indispensable. During viral infection, different transduction signaling pathways control various gene expressions that regulate several molecular and cellular avctivies required for the eradication of microorganisms and the regulation of inflammation [134]. The inflammation response must be regulated tightly because uncontrolled inflammation can lead to tissue damage. Hence, the ?-catenin signaling pathway is crucial due to its role in inhibiting the expression of several inflammation molecules during pathogen infection. In addition, ?-catenin signaling also plays an essential role in the processes of cell differentiation and stem cell self-renewal maintenance [141].
- The article is missing the explanation of the double sword effect of those cells which can be stimulators and also regenerative in certain conditions depending on the stimulus received by the cell from the homing environment. MSC side effect is lacking throughout the manuscript.
Response: This review article is focusing on MSCs role as an immunoregulatory, since MSCs role in regeneration has already been extensively reported. However, apparently we described MSCs as in regeneration and as immunomodulatory in brief on section 4.4, line 460–466, “Besides the repair and regenerative activities…….pro-inflammatory cytokine’ production [88]”. Furthermore, the immunoregulation properties also described briefly in section 6, which beta catenin is a signal that used by stem cells as regeneration and immunoregulation functions.
- More translational points should be added to the review article in order to enhance the clinical aspect of those cells and advances in the therapeutic protocols. Cell count/Ratio of infusion and timings in relation to the disease course is related to the amounts of mediators released and hence the amount of PGE 2 (concentration).
Response: In this review article, clinical aspects indeed are not explained in detail, because we are focusing to discuss the mechanisms of immunoregulation by MSCs. As for clinical aspects may need to further discussed in the next review article.
- English language needs to be revised.
Response: Following the suggestion, we submitted our manuscript to English editing service. The revised manuscript provided is the final result after English language editing process.

Reviewer 3 Report
1. "Nfκβ" should be replaced with "Nuclear factor kappa B (NF-κB)" , and "9.1.β-.catenin" should be rewritten as "9.1.β-catenin" in your manuscript.
2. Perhaps the fourth section of the manuscript, titled "4. Stem Cells," could be removed as it appears to be less relevant to the main topic.
3. One recommendation is to delve into the immune regulatory mechanism of allogeneic mesenchymal stem cells (MSCs) and their effects on the inflammatory response in the section of the manuscript titled "5. Mesenchymal Stem Cells.",And also, this section maybe appropriately simplified, because the immune regulation of PGE2 on exogenous mesenchymal stem cells is an important topic of discussion in the manuscript.
Author Response
Editor
International Journal of Molecular Sciences
03 April 2023
Subject: Revision and Resubmission of manuscript ijms-2233561
Dear Editor,
Thank you for the opportunity to revise our paper "PGE Promotes Immunoregulation of Exogenous Mesenchymal Stem Cells Through Activation of Wnt-β-catenin Signaling in Acute Respiratory Distress Syndrome Induced by Highly Pathogenic Influenza A Virus”. We are most grateful to you and the reviewers for the constructive comments and have made amendments accordingly.
I have included the editor comments immediately after this letter and responded to each of the points, indicating how we have addressed concerns and describing the changes we have made.
It is our sincere hope that the revised manuscript is now suitable for publication in International Journal of Molecular Sciences. We thank you for your continued interest in our research and look forward to hearing from you very soon.
Respectfully Yours,
Dr. Resti Yudhawati
Department of Pulmonology and Respiratory Medicine, Faculty of Medicine, Universitas Airlangga – Dr Soetomo General Academic Hospital
Jl. Prof. Dr Moestopo 6-8 Surabaya, 60286, Indonesia
Tel: +6289673691726
E-mail: resti-y-m@fk.unair.ac.id
Reviewers Comments:
Thank you for addressing the comment.
Reviewer 3 comments:
- “Nfκβ” should be replaced with “Nuclear factor kappa B (Nf-κβ)”, and “9.1.β-.catenin” should be rewritten as “9.1.β-catenin” in your manuscript.
Response: Following the suggestion, the amendments has been made accordingly.
- Perhaps the fourth section of the manuscript, titled “4. Stem Cells,” could be removed as it appears to be less relevant to the main topic.
Response: Following the suggestion and after long consideration, we decided to eliminate this section.
- One recommendation is to delve into the immune regulatory mechanism of allogeneic mesenchymal stem cells (MSCs) and their effects on the inflammatory response in the section of the manuscript titled “5. Mesenchymal Stem Cells”. And also, this section maybe appropriately simplified, because the immune regulation of PGE2 on exogenous mesenchymal stem cells is an important topic of discussion in the manuscript.
Response: The mechanisms of immunoregulation of allogeneic MSCs both via innate immune response and adaptive immune response is quite complex. This review article is focusing on the discussion of immunoregulatory mechanisms by MSCs via innate immune response, particularly by inhibiting Nf-κB through soluble factor released by allogeneic MSCs which will be explained in more detail in the next sub-topic.

Reviewer 4 Report
1. The reference are too old, and some related papers published in recent years are not cited in the paper.
2. In part 3, is the reference 35 is over-cited? The whole paragraph is cited the same reference.
3.Part 4: The introduction of stem cells. It is known that stem cells are a class of original cells with differentiation potential, including MSCs. Some functions of MSCs are similar to stem cells, but they are still different. I think it’s better focus on the title (MSCs) and delete Stem cell.
4. Part 8: TGF-β paragraph is less relevant to the topic.
5. The title: The PGE2 Promotes Immunoregulation of Exogenous Mes-enchymal Stem Cells Through Activation of Wnt-β-catenin in Part 9 only take three pages. The title should be reconsidered, maybe : Immunoregulation of Exogenous Mes-enchymal Stem Cells in Acute Respiratory Distress Syndrome Induced by Highly Pathogenic Influenza A Virus
Author Response
Editor
International Journal of Molecular Sciences
03 April 2023
Subject: Revision and Resubmission of manuscript ijms-2233561
Dear Editor,
Thank you for the opportunity to revise our paper "PGE Promotes Immunoregulation of Exogenous Mesenchymal Stem Cells Through Activation of Wnt-β-catenin Signaling in Acute Respiratory Distress Syndrome Induced by Highly Pathogenic Influenza A Virus”. We are most grateful to you and the reviewers for the constructive comments and have made amendments accordingly.
I have included the editor comments immediately after this letter and responded to each of the points, indicating how we have addressed concerns and describing the changes we have made.
It is our sincere hope that the revised manuscript is now suitable for publication in International Journal of Molecular Sciences. We thank you for your continued interest in our research and look forward to hearing from you very soon.
Respectfully Yours,
Dr. Resti Yudhawati
Department of Pulmonology and Respiratory Medicine, Faculty of Medicine, Universitas Airlangga – Dr Soetomo General Academic Hospital
Jl. Prof. Dr Moestopo 6-8 Surabaya, 60286, Indonesia
Tel: +6289673691726
E-mail: resti-y-m@fk.unair.ac.id
Reviewers Comments:
Thank you for addressing the comment.
Reviewer 4 comments:
- The reference are too old, and some related papers published in recent years are not cited in the paper
Response: Following the suggestion, we added more recent references in the paper.
- In part 3, is the reference 35 is over-cited? The whole paragraph is cited the same reference.
Response: Following the comment, we made revision on section 3.
- Part4: the introduction of stem cells. It is known that stem cells are a class of original cells with differentiation potential, including MSCs. Some functions of MSCs are similar to stem cells, but they are still different. I think it’s better focus on the title (MSCs) and delete Stem cell.
Response: Following the suggestion, we decided to remove section 4 after long consideration.
- Part 8: TGF-β is less relevant to the topic.
Response: Following the comment, we decided to remove the explanation of TGF-β, and only described it briefly in section 7 line 1048–1052.
- The title: The PGE2 Promotes Immunoregulation of Exogenous Mesenchymal Stem Cells Through Activation of Wnt-β-catenin in Part 9 only take three pages. The title should be reconsidered, maybe : Immunoregulation of Exogenous Mesenchymal Stem Cells in Acute Respiratory Distress Syndrome Induced by Highly Pathogenic Influenza A Virus.
Response: Following the suggestion, we added more explanation regarding PGE2 and Wnt-β-signaling in section 5 and 6. As for the title, we decided to revised it after long consideration to “PGE2 Promotes Exogenous MSCs Immunoregulation of Acute ARDS Induced by Highly Pathogenic Influenza A Through Activation of the Wnt-β-catenin Signaling Pathway”

Round 2
Reviewer 1 Report
Title PGE2 Promotes Exogenous MSCs Immunoregulation… sounds like you are administering PGE2. Better to say PGE2 Produced by Exogenous MSCs Improves Acute ARDS Induced… Introduction Acute respiratory distress syndrome (ARDS) is an acute respiratory failure that arises due to acute lung injury, which causes a complication [1,2]. ARDS is characterized by diffuse alveolar damage, which leads to severe… There are many complications. Line 75 function has NOT ?? drawn much attention as yet. Line 93 cytosolic retinol acid- RETINOIC Line 129 rec-ognize the complex with should be recognize Line 152 : mem-brane- should be membrane- Line 153 oli-gomerization should be oligomerization Line 161 Mac-rophages should be Macrophages Line 227 lin-ages should be lineages Line 234 MSCs were first introduced as an adherent, to a clonogenic, non-phagocytic Line 235-7 This was stated previously. Furthermore, these cells can be isolated from various other tissues, including adipose tissue, umbilical cord, placenta, and amniotic fluid [55]. PLEASE SPELL CHECK MANY of the references are not correct ! PLEASE READ YOUR MANUSCRIPT AND CHECK ALL REFERENCES Start checking at 50 and also check proper format such as Line 1132 et al. Weiss, D.J. et al. Stem Cells and Cell Therapies SOME REFs are used twice see Ortiz et al and Rojas et al

Author Response
Reviewers Comments:
Thank you for addressing the comment.
Reviewer 1 comments:
- Title PGE2 Promotes Exogenous MSCs Immunoregulation… sounds like you are administering PGE2. Better to say PGE2 Produced by Exogenous MSCs Improves Acute ARDS Induced…
Response: We are thanking the reviewer for suggesting the new title, however since the article is focusing on immunoregulation discussion, we believe it would be better to keep the “Immunoregulation” on the title, hence after considering the suggestion we revise the title to “PGE2 Produced by Exogenous MSCs Promotes Immunoregulation in Acute ARDS Induced by Highly Pathogenic Influenza A Through Activation of the Wnt-β-catenin Signaling Pathway”.
- Introduction Acute respiratory distress syndrome (ARDS) is an acute respiratory failure that arises due to acute lung injury, which causes a complication [1,2]. ARDS is characterized by diffuse alveolar damage, which leads to severe… There are many complications.
Response: Following the suggestion, the amendments has been made accordingly.
- Line 75 function has NOT ?? drawn much attention as yet. Line 93 cytosolic retinol acid- RETINOIC
Response: Following the suggestion, the amendments has been made accordingly.
- Line 129 rec-ognize the complex with should be recognize
Response: Following the suggestion, the amendments has been made accordingly.
- Line 152 : mem-brane- should be membrane-
Response: Following the suggestion, the amendments has been made accordingly.
- Line 153 oli-gomerization should be oligomerization
Response: Following the suggestion, the amendments has been made accordingly.
- Line 161 Mac-rophages should be Macrophages
Response: Following the suggestion, the amendments has been made accordingly.
- Line 227 lin-ages should be lineages
Response: Following the suggestion, the amendments has been made accordingly.
- Line 234 MSCs were first introduced as an adherent, to a clonogenic, non-phagocytic
Response: Following the suggestion, the amendments has been made accordingly.
- Line 235-7 This was stated previously. Furthermore, these cells can be isolated from various other tissues, including adipose tissue, umbilical cord, placenta, and amniotic fluid [55].
Response: Following the suggestion, the amendments has been made accordingly.
- PLEASE SPELL CHECK MANY of the references are not correct ! PLEASE READ YOUR MANUSCRIPT AND CHECK ALL REFERENCES Start checking at 50 and also check proper format such as Line 1132 et al. Weiss, D.J. et al. Stem Cells and Cell Therapies SOME REFs are used twice see Ortiz et al and Rojas et al
Response: We made the revision using Track Changes tool from Microsoft Word as requested by MDPI. Apparently, the revision on references and the narration of the article are made according to MDPI format in a very careful manner, however the tracking tool may have made the article become messy. Please make sure to click Accept All Changes and Stop Tracking in Review Tab, so the reference will not be messy nor appear twice.

Reviewer 2 Report
The authors had improved the manuscript yet some points are still needed to be considered;
Proinflammatory effects of MSC regarding ARDS (the topic of the manuscript) should be clearly considered. Similarly for the mode of infusion, pros and cons of the mode of infusion specified to the topic of the article. Indeed pulmonary trapping would be a beneficial effect rather than a disadvantage. Its role on mature and immature DCS should be considered which is directly related to the timing of MSC INFUSION. One of the key points in MSC infusion that it’s immunodulatory effect is trivial on already activated/ mature cells as IL2 activated NK CELLS and mature DC
human clinical trials should be mentioned more than ANIMAL trials and that what I meant by translational aspects should be considered discussing the pros and cons.
Author Response
Reviewers Comments:
Thank you for addressing the comment.
Reviewer 2 comments:
- The authors had improved the manuscript yet some points are still needed to be considered;
Proinflammatory effects of MSC regarding ARDS (the topic of the manuscript) should be clearly considered. Similarly for the mode of infusion, pros and cons of the mode of infusion specified to the topic of the article. Indeed pulmonary trapping would be a beneficial effect rather than a disadvantage. Its role on mature and immature DCS should be considered which is directly related to the timing of MSC INFUSION. One of the key points in MSC infusion that it’s immunodulatory effect is trivial on already activated/ mature cells as IL2 activated NK CELLS and mature DC
human clinical trials should be mentioned more than ANIMAL trials and that what I meant by translational aspects should be considered discussing the pros and cons.
Response: First of all we want to thank the reviewer for the suggestion. Investigation on MSCs’ role on ARDS cases induced by influenza A virus are widely conducted on animal model but still limited on human (written in Line 528–529 and 910–911). In addition, this article is discussing on the mechanisms of immunoregulation which may encourage further studies, particularly on human.
Regarding the mode of infusion, pros and cons of the mode of infusion, and the role of mature and immature DCs to the timing of MSC infusion, we believe it will be better to discuss it in another article with different title related to the immunoregulatory mechanisms of MSCs and their potential in clinical application in more detail. In our opinion, if we include this discussion into this very article, it will be too wide and confusing since our focus is on the one pathway of immunoregulation mechanisms which surely there are a lot of other pathways. In this article we are not discussing the clinical application on detail, however, the suggestion from the reviewer are very insightful and encourage us to write another article focusing on MSCs’ role in clinical application in more detail.

Reviewer 4 Report
The revision manuscripts have been made according to the modification suggestions. I think the manuscript meets the criteria for acceptance.
Author Response
Reviewers Comments:
Thank you for addressing the comment.
Reviewer 3 comments:
- The revision manuscripts have been made according to the modification suggestions. I think the manuscript meets the criteria for acceptance.
Response: We are most thankful to the reviewer for the constructive suggestions and the opportunity to accept our manuscript in this journal.